



# Changing sub-Arctic tundra vegetation upon permafrost degradation: impact on foliar mineral element cycling

Elisabeth Mauclet[1*], Yannick Agnan[1], Catherine Hirst[1], Arthur Monhonval[1], Benoît Pereira[1], Aubry Vandeuren[1], Maëlle Villani[1], Justin Ledman[2], Meghan Taylor[2], Briana L. Jasinski[2], Edward A. G. Schuur[2], Sophie Opfergelt[1]

[1]Earth and Life Institute, Université catholique de Louvain, Louvain-la-Neuve, Belgium
[2]Center for Ecosystem Science and Society, Northern Arizona University, Flagstaff, AZ, USA

*Correspondence to*: Elisabeth Mauclet (elisabeth.mauclet@uclouvain.be)

**Abstract.** Arctic warming and permafrost degradation are modifying northern ecosystems through changes in microtopography, soil water dynamics, nutrient availability, and vegetation succession. Upon permafrost degradation, the release of deep stores of nutrients such as nitrogen and phosphorus from newly thawed permafrost stimulates Arctic vegetation production. More specifically, wetter lowlands show an increase in sedges (as part of graminoids), whereas drier uplands favor shrub expansion. In turn, shifts in the composition of vegetation may influence local mineral element cycling through litter production. In this study, we evaluate the influence of permafrost degradation on mineral element foliar stocks and potential annual fluxes upon litterfall. We measured the foliar elemental composition (Al, Ca, Fe, K, Mn, P, S, Si, and Zn) on ~500 samples of typical tundra vegetation species from two contrasting Alaskan sites, i.e., under experimental (CiPEHR) and ambient (Gradient) warming. The foliar concentration of these mineral elements was species specific, with sedge leaves having relatively high Si concentration, and shrub leaves having relatively high Ca and Mn concentrations. Therefore, changes in the species biomass composition of the Arctic tundra in response to permafrost thaw are expected to be the main factors that dictate changes in elemental composition of foliar stocks and maximum potential foliar fluxes upon litterfall. We observed an increase in the mineral element foliar stocks and potential annual litterfall fluxes, with Si increasing with sedge expansion in wetter sites (CiPEHR), and Ca and Mn increasing with shrub expansion in drier sites (Gradient). Consequently, we expect that sedge and shrub expansion upon permafrost thaw will lead to changes in litter elemental composition, and affect nutrient cycling across the sub-Arctic tundra, with potential implications for further vegetation succession.

## 1 Introduction

Arctic and sub-Arctic ecosystems are typically characterized by persistently frozen ground (soil and/or rock), called permafrost, with a top active layer that seasonally thaws (Burn, 2013; French, 2013). The low soil temperatures and shallow active layers provide constraining environments for vegetation growth and development. With ongoing climate warming, increases in temperature, precipitation, and extreme event frequency (such as fires, riverine, or coastal flooding; IPCC, 2014) directly affect northern ecosystems and permafrost stability, resulting in changes in active layer thickness and water table





depth. Across the Arctic and sub-Arctic, permafrost degradation has been reshaping and modifying northern ecosystems through substantial changes in soil water content, soil nutrient availability, and vegetation succession (Jorgenson et al., 2006; Lloyd et al., 2003; Schuur and Mack, 2018; Yang et al., 2010).

Arctic and sub-Arctic tundra is mainly composed of vascular plant functional types such as graminoids, forbs, deciduous, and evergreen shrubs, but there is also a large contribution from non-vascular species such as mosses and lichens (Walker et al., 2005). The biomass distribution between these plant functional types responds to changing environmental conditions. For example, thermokarst processes (i.e., ground subsidence upon ground ice melting; Washburn, 1980) generate contrasted soil moisture regimes by altering ground surface topography and hydrology. Areas of subsidence accumulate water and may have

the water table near the soil surface, whereas nearby higher areas become drier (Schuur et al., 2007). In turn, these contrasting soil moisture conditions affect the nutrient availability, and the composition, distribution, and extent of tundra vegetation communities. Wetter soil conditions in subsided and poorly drained areas generally favor graminoid expansion (Jorgenson et al., 2015, 2001; van der Kolk et al., 2016) because of the ability of graminoids to grow in a wide range of soil moisture conditions and access nutrients at depth. Conversely, drier soil conditions upon warming drive an expansion in shrubs, called

shrubification (Jonsdottir et al., 2005; Shaver et al., 2001). In parallel, increase in air temperature influences nutrient availability for vegetation, by releasing nutrients from deeper mineral horizons upon permafrost thaw (Beermann et al., 2016; Keuper et al., 2017; Sistla et al., 2013), but also stimulating soil microbial activity, litter decomposition, and therefore nutrient mineralisation (Hobbie, 1996; Lavoie et al., 2011; Nadelhoffer et al., 1991; Rustad et al., 2001). In nutrient-limited ecosystems, such as Arctic and sub-Arctic tundra, an increase in nutrient availability (e.g., N and P) may strongly contribute to boost

vegetation biomass and productivity, and drive shifts in species composition (Dormann and Woodin, 2002; Jonasson et al., 1999; Mack et al., 2004; Nadelhoffer et al., 1991; Van Wijk and Williams, 2003; Walker et al., 2006).

Changes in Arctic and sub-Arctic tundra vegetation composition and productivity generate feedbacks on climate change, such as: (i) changing the surface energy balance, e.g., expanding plant cover in permafrost regions reduces the highly reflective snow cover, thereby amplifying global change (Chapin et al., 2005; Liston and Sturm, 2002); or (ii) modifying the delicate

balance between C uptake by plants through photosynthesis and C emissions by respiration, thereby affecting the net ecosystem C balance (Billings, 1987; Mack et al., 2004; Natali et al., 2019; Schuur et al., 2015, 2008; Shaver et al., 1992). Given these important feedbacks on climate change, efforts to model vegetation dynamics in these regions are ongoing. These efforts will improve the Earth system models used to predict the rate of climate change (Druel et al., 2019).

In the northern ecosystems, vegetation dynamics are largely dictated by the environmental conditions (such as low

temperatures and presence of permafrost), nutrient availability, and competition for resources. As pools of nutrients are limited by the presence of permafrost, nutrient cycling in the active layer provides a crucial source of nutrients. However, the parameterisation of vegetation dynamics and competition processes in the northern tundra lacks constraints on the impact of vegetation changes on nutrient cycling (Druel et al., 2019). Beside the major vegetation organic constituents (C, H, O), mineral nutrients can be sorted into essential macronutrients (e.g., N, P, K, Ca, S), micronutrients (e.g., Fe, Mn, Zn), and non-essential

nutrients (e.g., Si, Al) (Marschner, 2012). These elements support physiological functions in the vegetation, such as



biomolecule formation, processes regulation, and metabolic reactions (Marschner, 2012). As plants take up mineral nutrients from soils to sustain vital functions for growth and development, they play a central role in the biogeochemical cycling and flux of nutrients between soil and plant tissues through root uptake, seasonal leaf senescence, and litterfall. Litter decomposition rates are variable and depend on climatic conditions, soil characteristics, biochemistry of the produced litter,

microbial community composition, and the metabolism of decomposer organisms (Brovkin et al., 2012; Cornelissen et al., 2007; Cornwell et al., 2008; Dorrepaal et al., 2005). In the Arctic and sub-Arctic tundra, the leaf decomposition rate is largely dependent on the plant functional type: higher for sedges compared to shrubs (Hobbie, 1996), and higher for deciduous shrubs compared to evergreen shrubs (Parker et al., 2018). Therefore, shifts in vegetation (e.g., sedge or shrub expansion) may accelerate or slow the litter decomposition, and influence the recycling of mineral nutrients.

In the context of plant mineral nutrition (Chapin, 1980; Marschner, 2012), Arctic tundra major plant functional types show variabilities in element uptake, accumulation, and cycling (Aerts and Chapin, 1999; Chapin, 1980; Chapin et al., 1996; Cornelissen et al., 1996; Grime et al., 1997; Hobbie and Gough, 2002). More specifically, many studies across the Arctic have focused on N and P, because of their large requirement for plant development and their limited availability for plants in permafrost regions (Chapin et al., 1995; Finger et al., 2016; Hewitt et al., 2019; Jonasson, 1983; Keuper et al., 2012; Mack et

al., 2010; Salmon et al., 2016; Shaver and Chapin, 1991; Zamin et al., 2014), and a few on Si (Carey et al., 2019, 2017) and K, Ca, Mg (Chapin III et al., 1980; Jonasson, 1983). To our knowledge, less attention has been paid to the influence of Arctic warming on the plant tissue accumulation of these mineral nutrients and others (i.e., Al, Ca, Fe, K, Ca, Mn, S, Si, and Zn), and to their associated annual return to the soil upon litterfall. According to their physiological and biogeochemical characteristics, Arctic and sub-Arctic tundra vegetation species display diverse strategies for mineral elements uptake and accumulation into

their tissues. Therefore, there is a need to investigate the variability in mineral element distribution among the tundra vegetation species, and the cycling rate of these elements by species, information required to improve simulations of vegetation dynamics in the changing northern ecosystems (Druel et al., 2019) and to feed emerging frameworks for spatial ecosystem ecology (Leroux et al., 2017).

In this study, we evaluate how changes in vegetation biomass composition, driven by permafrost degradation, may influence

the mineral element fluxes between vegetation tissues and soil litter in a typical sub-Arctic moist acidic tundra. Specifically, we considered the individual contributions of typical vegetation species from Arctic and sub-Arctic tundra to mineral elements stocks and fluxes. Our study investigated tundra species-specific foliar elemental composition to address the following research question: what is the influence of a shift in vegetation biomass composition on the total mineral element foliar stocks at site scale? Based on these data, we investigate the influence of a shift in vegetation biomass composition on the maximum potential

fluxes of mineral elements from plant leaves to soil litter upon annual leaf senescence and litterfall. To this end, we relied on two contrasted study sites, covering graminoid- and shrub-dominated tundra: an experimental permafrost warming site and a natural thermokarst gradient ranging minimal to extensive permafrost degradation, which both show changes in vegetation species composition with thawed permafrost.



## 2 Material and methods

### 1.2 Study site

The study was conducted in the northern foothills of the Alaska Range in sub-Arctic tundra, within the Eight Mile Lake (EML) watershed in Healy, Alaska, USA (63°52'42N, 149°15'12W; Schuur et al., 2009). The site is underlain by degrading permafrost in the discontinuous permafrost zone (Natali et al., 2011; Osterkamp et al., 2009). Climate is characterized by a mean annual air temperature of –1 °C (1977-2015), with mean monthly temperatures ranging from –16 °C in December to +15

°C in July (Healy and McKinley Stations, Western Regional Climate Center, and National Oceanic and Atmospheric Administration National Centers for Environmental Information [NOAA]). Average annual precipitation is 381 mm (2007-2017; https://www.usclimatedata.com/climate/healy/alaska). Soils are characterized by a 35 to 55 cm thick organic layer (>20% of organic C content) at the surface, lying above a cryoturbated mineral soil (5-20% of organic C) composed of glacial till and loess parent material (Hicks Pries et al., 2012; Osterkamp et al., 2009; Vogel et al., 2009). Soil pH$_{water}$ fluctuates

between 3 and 5 throughout the profile (Bracho et al., 2016; Osterkamp et al., 2009). The site is located on moist acidic tundra, with a dominance of tussock-forming sedges, such as *Eriophorum vaginatum* L. and *Carex bigelowii* Torr. ex Schwein, evergreen shrubs (e.g., *Andromeda polifolia* L., *Rhododendron tomentosum* Harmaja, *Vaccinium vitis-idaea* L., and *Empetrum nigrum* L.), deciduous shrubs (e.g., *Vaccinium uliginosum* L. and *Betula nana* L.), and forbs (e.g., *Rubus chamaemorus* L.). Non-vascular plant cover is dominated by mosses (mainly *Sphagnum* spp., *Dicranum* spp., and feather mosses including

*Hylocomium splendens* and *Pleurozium schreberi*,) and lichen species (e.g., *Nephroma* spp., *Cladonia* spp., and *Flavocetraria cucullata*) (Deane-Coe et al., 2015; Natali et al., 2012; Schuur et al., 2007).

Within the EML watershed, an experimental warming project called CiPEHR (Carbon in Permafrost Experimental Heating Research) was established in 2008. The site was established on a relatively well-drained gentle (~3°) north-east-facing slope. The experimental set up gathered 48 plots distributed equally into four different artificial warming treatments that are applied

on top of a natural background of warming. Treatments will be referred to as follows: control (12 plots with no artificial warming treatment), summer warming (12 plots subjected to summer air warming only), winter warming (12 plots subjected to winter soil warming only), and annual warming (12 plots subjected to both: summer and winter warming). Winter warming consisted of a soil warming treatment, that used six snow fence replicates (plots 1.5 m tall × 8 m long, n = 6) to increase winter snow depth and insulate the soils from extremely cold air temperatures. The extra snow was removed every spring to avoid

excess soil moisture due to snowmelt between winter warming and control plots (Natali et al., 2011). Summer warming consisted of an air warming treatment using Open Top Chambers, set out during the snow-free period (from May to September). Further description of the experimental design can be found in Natali et al. (2011). When CiPEHR was installed in 2008, the maximum seasonal extent of permafrost thaw depth was ~50 cm and this has since increased at a rate of 2 cm a$^{-1}$ in control areas and 6 cm a$^{-1}$ in soil warming areas (Mauritz et al., 2017). CiPEHR site displayed shallower water tables upon

permafrost degradation and soil subsidence processes between 2009 and 2017, resulting in wetter soil conditions in 2017 than in 2009 (Rodenhizer et al., 2020).





Additionally, a natural gradient in permafrost thaw and thermokarst formation has developed at EML since the mid to late 1980s (Schuur et al., 2007), and has been defined as the Gradient site. The site is located on a gentle (< 5°) north-facing slope (Osterkamp and Romanovsky, 1999; Schuur et al., 2007) and has been monitored since 1990 to follow the impact of permafrost

degradation on ground subsidence, thaw depth and water table depth (Schuur et al., 2009). Recent Bonanza Creek LTER database (https://www.lter.uaf.edu/data) reports contrasted maximum active layer thickness and water table depth for Minimal, Moderate, and Extensive areas (measurements in 2019), which reflect different stages of permafrost degradation, resulting in drier soil conditions at Moderate and Extensive than at Minimal area. Since the beginning of the monitoring, Minimal thaw area showed less ground subsidence and little-disturbed moist acidic tundra, and is dominated by the sedge *E. vaginatum* and

*Sphagnum* spp. mosses, coexisting with evergreen and deciduous shrubs (Jasinski, 2018; Schuur et al., 2007). The Moderate thaw area displays isolated areas of ground subsidence (Schuur et al., 2007) and remains dominated by the sedges *E. vaginatum*, with a lower moss cover than the Minimal area (Jasinski et al., 2018). The Extensive thaw area is characterised by large-scale ground subsidence leading to an undulating surface micro-topography and high heterogeneity in active layer depths and water table depths, with water accumulation in lower areas and drier soil conditions in higher areas. Since the start of the thermokarst

development, vegetation cover changed with evergreen and deciduous shrubs (as *V. uliginosum* and *R. tomentosum*), and forbs (as *R. chamaemorus*) being dominant at the expense of tussock forming sedges (Jasinski et al., 2018; Schuur et al., 2007).

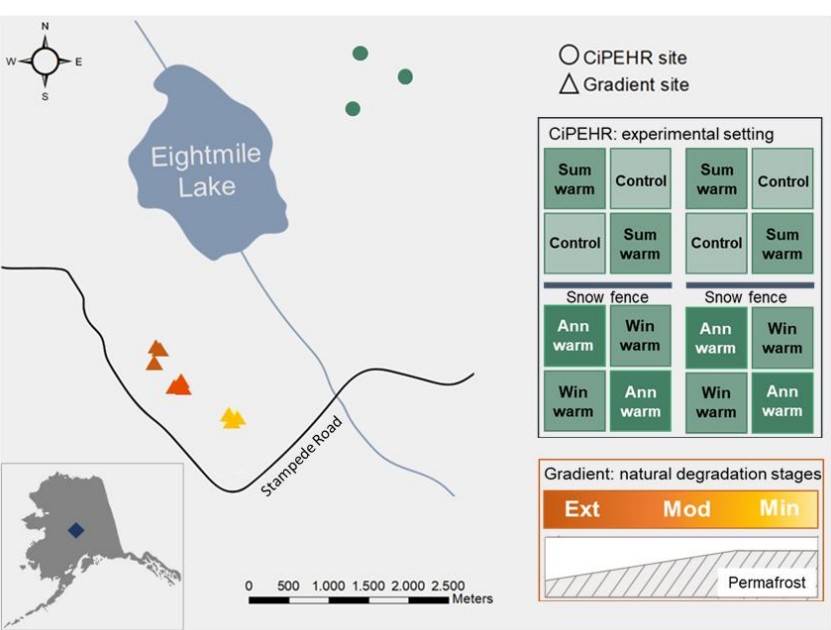

**Figure 1: Study site at Eight Mile Lake, in Central Alaska, USA. CiPEHR site (Carbon in Permafrost Experimental Heating Research) is an experimental site with 4 treatments: control, summer air warming (Sum warm) with open top chamber settings,**
**winter soil warming (Win warm) with snow fences that insulate the soil upon snow accumulation, and annual warming (Ann warm) with both, air and soil warming. Gradient site is a natural thermokarst gradient composed of three contrasted areas in terms of permafrost degradation, with minimal (Min), moderate (Mod), and extensive (Ext) permafrost degradation. Source: Esri, HERE, Garmin, OpenStreetMap contributors and GIS user community.**

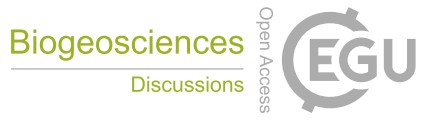

### 1.2 Sampling method

At CiPEHR site, we selected five of the most abundant vascular species from the moist acidic tundra: *E. vaginatum*, *B. nana*, *V. uliginosum*, *R. chamaemorus*, and *V. vitis-idaea*. The sampling was performed on most of the 48 plots (Table S.1) at peak growing season in July 2009 (before the experimental warming start) and in July 2017 (after 8 years of experimental warming). The sampling method was similar between 2009 and 2017, and involved the collection of fully formed green leaves from the current year's growth over an area of ~1 m$^2$. At Gradient site, we selected seven of the most abundant vascular species: *E.*

*vaginatum*, *C. bigelowii*, *B. nana*, *V. uliginosum*, *R. chamaemorus*, *R. tomentosum*, and *V. vitis-idaea*. We sampled fully formed leaves of each vascular species over an area of ~5 m$^2$ at three plots in each thaw area (Minimal, Moderate, and Extensive area; Table S.1). We also sampled moss (*Sphagnum* sp., *Brachytecium* sp., and *Dicranum* sp.) and lichen species (*Nephroma* sp., *Cladonia* sp., *Flavocetraria cucullata*). The sampling was performed between the end of August and mid-September 2019 (i.e., at the late season period). For both sites, leaf samples were dried at 60 °C and ground (Jasinski, 2018; Natali et al., 2011;

Schuur et al., 2007).

### 1.3 Mineral elemental analysis

Ex-situ mineral element concentrations (Al, Ca, Fe, K, Mn, P, S, Si, and Zn) were determined on 506 leaf samples (437 at CiPEHR (2009-2017) and 69 at Gradient in 2019) using the non-destructive portable X-ray fluorescence (pXRF) device Niton xl3t Goldd+ (Thermo Fisher Scientific, Waltham, USA). For the measurement, we fixed a transparent film (prolene 4 µm) at

the base of a circular plastic cap, in which we placed the dried and ground foliar sample powder in order to reach ~1 cm of sample thickness. We conducted the analyses in laboratory conditions, using a lead stand to protect the operator from X-rays emission, and total time of measurement was 90 seconds. The pXRF-measured concentrations were calibrated with another accurate wet chemistry method, using measurements by inductively coupled plasma optical emission spectroscopy (ICP-OES, iCAP 6500 ThermoFisher Scientific, Waltham, USA) after alkaline fusion, for 90 foliar samples (more information in Text

S.1 and Fig. S.1). Corrected pXRF foliar concentrations of every species on every site are summarized in Mauclet et al. (2021a, 2021b). Raw data were acquired from averaging three individual pXRF measurements and used to get means and standard deviations of site-specific foliar concentrations.

### 1.4 Data treatment

For both sites (CiPEHR and Gradient), aboveground biomass datasets of sampled vegetation species were available: for

CiPEHR in 2009 and 2017 (Taylor et al., 2017) and for Gradient site in 2017 (Jasinski et al., 2018). Aboveground biomasses were determined by a non-destructive point frame method (Natali et al., 2012; Schuur et al., 2007; Shaver et al., 2001), in late July. To convert aboveground biomass of vascular species to foliar biomass, we applied species-specific ratios (r) that were determined at the study site of Eight Mile Lake (Alaska, USA) (Table S.2; Salmon et al., 2016; Schuur and Crummer, 2009).



For moss and lichen species, we assumed a ratio between foliar biomass and aboveground biomass of 1. We calculated foliar
biomasses (FB) with the following equation:

$$FB = AB \times r \qquad (1)$$

With FB, foliar biomass (g m$^{-2}$), AB, the aboveground biomass (g m$^{-2}$), and r, the ratio between foliar biomass and
aboveground biomass (unitless).

The estimation of mineral element foliar stocks (FS) was then calculated using the following equation:

$$FS = FC \times FB \times \frac{1}{1000} \qquad (2)$$

With FS, mineral element foliar stock (mg m$^{-2}$), FC, elemental foliar concentration (mg kg$^{-1}$), and FB, foliar biomass (g m$^{-2}$).
We further calculated standard deviations ($\Delta$ FS) in order to assess the heterogeneity of mineral element foliar stocks among
species and across the different sites with the equation:

$$\Delta FS = [\Delta FC \times FB + FC \times \Delta FB] \times \frac{1}{1000} \qquad (3)$$

At an annual time-scale and considering productivity ratios that were integrated over several years (Schuur and Crummer,
2009), we assumed that sub-Arctic tundra reached its equilibrium and the annually produced biomass (net primary productivity,
NPP) is equivalent to the senescing biomass (biomass returning to the soil litter). Therefore, the estimation of annual foliar
fluxes (FF) of mineral elements from standing leaf to leaf litter was calculated using the following equation:

$$FF = FC \times fNPP \times \frac{1}{1000} \qquad (4)$$

With FF, mineral element foliar flux (mg m$^{-2}$ a$^{-1}$), FC, elemental foliar concentration (mg kg$^{-1}$) and fNPP, foliar NPP (g m$^{-2}$
a$^{-1}$). Foliar NPP (fNPP) were assessed from site-specific biomass data (CiPEHR: Taylor et al., 2018 and Gradient: Jasinski et
al., 2018), and from ratios of species-specific foliar production established by Schuur et al. (2007) (Table S.4). We calculated
standard deviations of annual foliar fluxes ($\Delta$ FF) using the following equation:

$$\Delta FF = \Delta FC \times fNPP \times \frac{1}{1000} \qquad (5)$$

At CiPEHR, data of elemental foliar concentration (FC) and foliar biomass (FB) were available for the five vascular plant
species, and most of the 48 plots subjected to artificial warming treatment, in 2009 and 2017. Therefore, we were able to
calculate the mineral element foliar stocks (FS, Eq. (2)) and annual fluxes (FF, Eq. (4)) of each plant species, warming
treatment, and year (2009 and 2017). Our results present averages of the plant-specific FS and FF from plots belonging to the
same treatment and year. At Gradient, given the different experimental design, we first averaged FB, fNPP, and FC for each
(vascular and non-vascular) species and for each area of permafrost thaw gradient (Minimal, Moderate, or Extensive). Using
these averaged factors (FB, fNPP, and FC), we estimated the FS (Eq. (2)) and FF (Eq. (4)) for the three thaw gradient areas.





All statistics were performed using R 4.0.2 (R Core Team, 2020) and plots using the *ggplot2* package (Wickham, 2016). At CiPEHR, the significant differences in mineral element stocks and fluxes between modalities were evaluated using a mixed model analysis (*lme4* package, Bates et al., 2015), with individual plots as random factors, year and treatments (control, summer warming, winter warming, and annual warming) as covariates (Table S.5). The model includes interactions (i) between treatments (summer × winter), and (ii) between treatments and years (2009 and 2017). At Gradient, we tested the influence of

the permafrost degradation degree (minimal, moderate, and extensive) on elemental foliar stocks and annual fluxes with a one-way analysis of variance (ANOVA) and a Tukey's honestly significant difference (HSD) post-hoc test.

## 3 Results and discussion

### 3.1 Foliar mineral element concentration at CiPEHR and Gradient

At the experimental warming site (CiPEHR), the average elemental foliar concentrations of the five vascular plant species

generally decreased for all treatments in the order Si > K > Ca > Al > P > Mn > S > Fe > Zn, in 2009 (Fig. 2a) and in 2017 (Fig. 2b). Plant functional types showed contrasting nutrient foliar concentrations, according to their physiology and growing strategies (Aerts et al., 1999). For example, high Si concentrations are common in sedge tissues, as observed in *E. vaginatum*, and can be attributed to strategies of Si root uptake (Ma and Takahashi, 2002). Silicon is known to promote better access to light and oxygen by strengthening the leaves (Hodson et al., 2005; Ma and Takahashi, 2002; Quigley et al., 2017). Here, we

observed an increase in Si foliar concentration between 2009 and 2017. Conversely, the low Ca foliar concentration in sedges (here, *E. vaginatum*) compared to shrubs and forbs results from different Ca requirements for growth between monocotyledons and dicotyledons (Loneragan and Snowball, 1969; Marschner, 2012). A low demand in Ca for monocotyledons may reflect a low capacity for Ca uptake (through $Ca^{2+}$ binding sites in the cell walls; White and Broadley, 2003), storage, or management into cells.

At the thermokarst gradient site (Gradient), elemental foliar concentrations decreased in the order K > Ca > Si > P > Mn > Al > S > Fe > Zn (Fig. 2c). Additional species were included, i.e., the sedge *C. bigelowii*, the evergreen shrub *R. tomentosum*, and non-vascular moss and lichen species. *C. bigelowii* displayed high Si foliar concentration that corroborates the sedge affinity for Si. Moreover, mosses and lichens also displayed singularities in nutrient concentrations compared to vascular species, with high concentrations in Si (up to 6 g kg$^{-1}$), Al (up to 1.5 g kg$^{-1}$), and Fe (up to 0.6 g kg$^{-1}$), potentially induced by the presence

of local minerogenic dust deposition (Reimann et al., 2001).

We observed that mineral element foliar concentrations were mostly specific to plant functional types, reflecting different vegetation uptake and storage strategies. Therefore, foliar elemental composition of vegetation species constitutes a key and reliable parameter to evaluate how the mineral element foliar stock may evolve upon changing vegetation species abundance and biomass.

**Biogeosciences** Open Access
Discussions
EGU




**Figure 2: Mineral element concentrations (Si, K, Ca, Al, P, Mn, S, Fe, and Zn) in foliage at (a) the Carbon in Permafrost Experimental Heating Research (CiPEHR; average over all warming treatments) in 2009 and (b) 2017, and (c) the thermokarst gradient site (Gradient; average over all degradation stages) in 2019. Vegetation species are sorted by plant functional types: deciduous shrubs and forbs (green), evergreen shrubs (blue), sedges (yellow), lichens and mosses (red). Error bars represent**
**standard deviations.**



### 3.2 CiPEHR: influence of experimental warming on mineral element foliar stocks and fluxes

This section investigates the influence of different warming treatments (control, summer warming, winter warming, and annual warming) on mineral element foliar stocks and annual fluxes from plant leaves to soil litter (online data: Mauclet et al., 2021a).

### 3.2.1 Influence of experimental warming on mineral element foliar stocks

At CiPEHR, the average elemental foliar stocks in 2009 varied from 800-900 mg m$^{-2}$ for K to 5-6 mg m$^{-2}$ for Zn (as K > Si > Ca > P > Al > Mn > S > Fe > Zn), whereas the average elemental foliar stocks in 2017 ranged from 2000-2500 mg m$^{-2}$ for Si to 10 mg m$^{-2}$ for Zn (as illustrated for Control site in Fig. 3a). Between 2009 and 2017, the increase in Al, P, S, Fe, and Zn foliar stocks at Control site was significant (p < 0.05; Table S.5), with at least 40% increase between 2009 and 2017 (Fig. 3a), and a strong variation in the species contribution to the global foliar stocks. These changes in elemental foliar stocks were

mainly biomass-driven, with the massive sedge expansion over time (Taylor et al., 2018). Additionally, the foliar elemental composition of vegetation species influenced patterns of change in elemental foliar stocks over time (e.g., no significant increase in Ca and Mn foliar stocks, compared to other nutrients: Fig. 3a). While in 2009, the sedge (*E. vaginatum*) contributed nearly 50% to the nutrient foliar stocks, that contribution went up to ~85% for some nutrient foliar stocks (i.e., Si and P) in 2017. The increase in *E. vaginatum* biomass over time across the site (1.7- to 3- times higher in 2017 than in 2009; Taylor et

al., 2018) occurred in areas of soil subsidence with wetter soil conditions, driven by permafrost degradation between 2009 and 2017 (Rodenhizer et al., 2020). In permafrost-affected landscapes, graminoid (including sedge) expansion seems to evolve towards wetter soil conditions (van der Kolk et al., 2016), with evidence provided here for a subsequent increase in the sedge contribution to the nutrient foliar stock.

The effect of the four artificial warming treatments on foliar stocks was evaluated in 2009 and in 2017 for Si, Ca, and Mn (Fig.
4a-c-e), and for K, Zn, S, P, Fe, and Al (Fig. S.2). In 2009 (before any experimental warming), as in 2017 (after eight years of experimental warming), mineral element foliar stocks showed no significant difference (p > 0.05) between the warming treatments and the control site, despite an increasing degree of permafrost degradation between the four warming treatments in terms of subsidence and rising water table (control < summer warming < winter warming < annual warming; Rodenhizer et al., 2020). Only time had a positive effect on mineral element foliar stocks, with total foliar stocks being 1.5- to 5-times higher

in 2017 than in 2009 (p < 0.05, for almost every mineral element, except for Ca and Mn). More specifically, total Si foliar stocks ranged from 550 to 650 mg m$^{-2}$ in 2009, before reaching 3- to 4-times higher values in 2017 (Fig. 4a). This massive increase in Si foliar stocks was mainly biomass-driven, i.e., arose from the large increase in *E. vaginatum* biomass between 2009 and 2017 (Taylor et al., 2018). To a lower extent, this change in Si foliar stocks was also concentration-driven, i.e., emphasized by the increase in Si concentration into sedge foliar tissues between 2009 and 2017 (Fig. 2a-b). This resulted in a

large increase in sedge contribution to Si foliar stocks (from ~40% in 2009 to ~85% in 2017). Silicon foliar stocks in *B. nana*, *V. uliginosum*, *R. chamaemorus*, and *V. vitis-idaea* remained relatively constant between 2009 and 2017, because their respective biomasses were similar over time. Furthermore, total foliar stocks of Ca and Mn (Fig. 4c-e) showed lower increases



than Si between 2009 and 2017, with increasing factors ranging around 1.3. These contrasted scenarios are mainly explained
by the lower Ca and Mn foliar concentrations in sedges, compared to shrub and forb species (Fig. 2a-b). While the sedge (i.e.,

*E. vaginatum*) was leading the net increase in total biomass between 2009 and 2017, shrub and forb species (i.e., *B. nana*, *V.
uliginosum*, *R. chamaemorus*, and *V. vitis-idaea*) displayed very low increase or decrease in their biomasses (Taylor et al.,
2018). Consequently, mineral elements that accumulate in higher concentrations into shrubs and forbs than into sedges (such
as Ca and Mn) displayed lower increase in their total foliar stocks between 2009 and 2017. At CiPEHR, foliar elemental
composition of *E. vaginatum* mainly influenced the evolution of the mineral element foliar stocks. It follows that the mineral

elements concentrated in *E. vaginatum* foliar tissues (such as Si; Fig. 2a-b) are those showing high increase in their foliar
stocks with time, as a result of the large sedge expansion upon the artificial warming. Conversely, mineral elements more
concentrated into shrub than sedge foliar tissues (such as Ca and Mn; Fig. 2a-b) showed much increase in their mineral element
foliar stocks upon warming, because of the shrub biomass temporal stability, and the more restricted sedge contribution to
their foliar stocks.

The high variability associated with average elemental foliar stocks (Fig. 4a-c-e and Fig. S.2) was mostly explained by the
important heterogeneity in species foliar biomasses among sites from similar treatment. On average, foliar biomasses had
relative standard deviations 2- to 5-times higher than foliar concentrations (Table S.3) (Mauclet et al., 2021a). This highlights
the large influence of the heterogeneity in vegetation biomass distribution on the foliar stock standard deviation.

It should be noticed that mineral element foliar stock estimates at CiPEHR only included contribution from the five most

abundant vascular plant species on site, covering ~55% of the total foliar biomass in 2009, and ~72 % in 2017. Foliar stock
estimates did not include non-vascular species (i.e., mosses and lichens), accounting for ~15 and ~13%, respectively, of the
foliar biomass in 2009, and for ~7 and ~5%, respectively, in 2017. Therefore, the results (Fig. 4a-c-e and Fig.S.2) provide
lower estimates compared to the actual total foliar stocks at CiPEHR.

### 3.2.2 Influence of experimental warming on potential mineral element foliar fluxes

Changes in mineral element foliar stocks as a result of changing vegetation composition have important implications for
changes in elemental composition of foliar litterfall fluxes. We quantified potential annual elemental foliar fluxes from
deciduous species (sedges, deciduous shrubs, and forbs), that lose their leaves each season. We did not include evergreen
species because of the complexity of their leaf and tissues lifespan and senescence dynamics (Park et al., 2020; Vitt, 2007).

As we based our foliar litterfall flux calculations on elemental foliar concentrations measured in July (at the peak of the growing

season), some mobile nutrients may translocate from mature leaves to storage organs or growing tissues throughout the season
(i.e., between the sampling period in July and the actual foliar litterfall fluxes in August). This process of nutrient
retranslocation occurs for some mineral elements (here, P and K; Berendse and Jonasson, 2012; Chapin and Shaver, 1989;
Chapin III et al., 1980; Jonasson and Chapin, 1985), into certain plant species, and upon specific plant nutritional status (Chapin
III et al., 1980; Maillard et al., 2015; Marschner, 2008; White, 2012). Therefore, we may overestimate the total annual foliar

fluxes of K and P (Fig. S.4). Conversely, our estimates for annual foliar fluxes of non-mobile nutrients, such as Ca and Mn





(Chapin III et al., 1980; Maillard et al., 2015; Marschner, 2008; White, 2012) are representative of the actual foliar fluxes (Fig. 6c-e). Finally, only few studies reported data on Si, S, Zn, Al, and Fe mobility. While mobility of S, Zn, and Fe into plant tissues seems to depend on plant-specific prerequisites (Maillard et al., 2015; Marschner, 2008), and plant deficiency (Shi et al., 2011), Si tends to accumulate in foliar tissues through the growing season (Shi et al., 2011). For these reasons, we will
further talk about "maximum potential foliar flux" of mineral elements upon litterfall.

In 2009, potential mineral element foliar fluxes varied from 680 mg $m^{-2}$ $a^{-1}$ for K to 5 mg $m^{-2}$ $a^{-1}$ for Zn, and were 2- to 10-times higher in 2017, as illustrated for the Control site (Fig. 5a). This significant increase in potential elemental foliar fluxes over time (p < 0.05, Table S.5) was mainly biomass-driven, and explained by the large increase in *E. vaginatum* foliar biomass (Table S.3), and productivity (Table S.4). Consequently, the sedge (*E. vaginatum*) foliar elemental composition largely
influenced the patterns of change in elemental foliar fluxes over time (Fig. 3a), and its contribution to the total annual foliar fluxes went from 40-70% in 2009 (for Ca and P, respectively), up to 90% in 2017 (for Si, P, and Al). To a lower extent, some changes in the potential elemental foliar fluxes (i.e., Si and Al) were also concentration-driven, i.e., emphasized by the increase in Si and Al concentration into sedge foliar tissues (Fig. 2a-b). Both mineral elements (Si and Al) showed the highest increases in potential elemental foliar flux (mean of 5- and 10-times, respectively). The overall shrub and forb contribution to the
potential annual foliar fluxes decreased over time.

The effect of the four artificial warming treatments on the potential annual foliar fluxes was evaluated in 2009 and in 2017 for Si, Ca, and Mn (Fig. 6a-c-e), and for K, Zn, S, P, Fe, and Al (Fig. S.4). In 2009, some mineral elements (Ca and Mn: Fig. 6c-e; K, Fe, S, and Zn: Fig. S.4) showed differences in their potential elemental foliar fluxes. As no warming treatment started yet, we attribute these differences to a background level of heterogeneity. In 2017 (after eight years of experimental warming),
few mineral elements also showed differences in their potential foliar fluxes according to the warming treatment, e.g., potential K and Fe foliar fluxes were significantly higher (p < 0.05) upon winter warming than control (30% and 20% higher, respectively; Fig. S.4), while potential K foliar flux was significantly lower (p < 0.05) upon summer warming than control (15% lower; Fig. S.4). However, potential mineral element fluxes from plant leaves to soil litter mainly showed an overall significant increase over time (between 1.5 to 10-times), that is much more pronounced than the variability between the four
treatments. More specifically, potential total foliar fluxes of Si increased more than 5-times between 2009 and 2017, i.e., from 320-360 mg $m^{-2}$ $a^{-1}$ in 2009 to 1650-2300 mg $m^{-2}$ $a^{-1}$ in 2017 (Fig. 6a). The main contributors to annual Si foliar fluxes were *E. vaginatum* (~50% in 2009 and ~90% in 2017), and *V. uliginosum* (15-20% in 2009 and 6-10% in 2017). Furthermore, potential annual Ca and Mn foliar fluxes (Fig. 6c-e) only increased from 1.3- to 2.3-times between 2009 and 2017. Potential Ca foliar fluxes increased from 200-285 mg $m^{-2}$ $a^{-1}$ in 2009 to 380-390 mg $m^{-2}$ $a^{-1}$ in 2017, and potential Mn foliar fluxes
increased from ~60-85 mg $m^{-2}$ $a^{-1}$ in 2009 to 110-130 mg $m^{-2}$ $a^{-1}$ in 2017. Despite the low contribution of *B. nana* to potential annual Ca and Mn foliar fluxes, shrubs (i.e., *V. uliginosum*) and forbs (i.e., *R. chamaemorus*) contribution was more important to Ca and Mn than to potential annual Si foliar fluxes. It should be noted that the lower standard deviations for annual foliar flux estimates (compared to foliar stock estimates) may be underestimated because we did not integrate standard deviation of the foliar NPP.





Overall, patterns of change in mineral element foliar stocks, and thereby in potential foliar fluxes between vegetation tissues and soil litter, were mostly biomass-driven with the sedge (*E. vaginatum*) expansion across the site. At CiPEHR site, the warming experiment generated soil subsidence and wetter soil conditions upon rising water table depths (Rodenhizer et al., 2020), which created favourable conditions for sedge development, at the expense of woody shrubs (van der Kolk et al., 2016). Consequently, elements highly concentrated in sedge leaves (here, *E. vaginatum*), such as Si, P, or Fe (Fig. 2a-b), displayed

important change in their foliar stocks (Fig. 4a and Fig. S.2), and thereby in their annual foliar fluxes, between 2009 and 2017 (Fig. 6a and Fig. S.4). On the other hand, elements highly concentrated in deciduous shrubs and/or forbs, such as Ca and Mn (Fig. 2a-b), displayed a more mitigated increase in their foliar stocks (Fig. 4c-e), and thereby in their potential annual foliar fluxes, between 2009 and 2017 (Fig. 6c-e).

**3.3 Gradient: influence of natural permafrost degradation on mineral element foliar stocks and fluxes**

In this section, we focus on seven vascular species and two groups of non-vascular species (mosses and lichens) to assess the influence of drier soil conditions upon natural permafrost degradation on mineral element foliar stocks and fluxes from plant leaves to soil (online data: Mauclet et al., 2021b).

**3.3.1 Influence of permafrost degradation on mineral element foliar stocks**

At Gradient site, total mineral element foliar stocks varied from ~1350 mg m$^{-2}$ for K to ~13 mg m$^{-2}$ for Zn (as illustrated for

the Minimal area; Fig. 3b). The foliar elemental composition (Fig. 2c) and biomass (Jasinski et al., 2018) of each plant functional type (i.e., mosses, sedges, shrubs, and forbs) explained their specific contribution to the mineral element foliar stocks. Mosses contributed massively to the total elemental foliar stocks (from 30% for Zn and K, up to more than 60% for Si, Al, and Fe), while sedge contribution to mineral element foliar stocks ranged from 7% for Fe and 40% for Mn. Shrubs and forbs accounted for 5% of Si and Fe foliar stock, and for ~40% of Ca and Mn foliar stocks.

At Gradient site, we collected leaf samples late in the growing season, when retranslocation of nutrients already occurred from mature leaves to storage organs or growing tissues. As discussed in section 3.2.2, we may therefore underestimate the mineral element foliar stocks (Fig. S.3) of mobile mineral elements such as P and K (Berendse and Jonasson, 2012; Chapin III et al., 1980; Maillard et al., 2015; Marschner, 2008). For Ca and Mn, that are non-mobile nutrients in plant tissues (Chapin III et al., 1980; Maillard et al., 2015; Marschner, 2008; White, 2012), the foliar stock estimations (Fig. 6d-f) can be considered as

representative of the actual foliar stocks. For Si, S, Zn, Al, and Fe, for which the mobility remains poorly documented, the foliar stocks of some elements may also be underestimated to a lesser extent.

Areas spanning the thermokarst development at Gradient site (Minimal, Moderate, and Extensive) displayed differences in their mineral element foliar stocks, and in the species contribution to these foliar stocks (Si, Ca, and Mn in Fig. 4b-d-f and K, Al, P, S, Fe, and Zn in Fig. S.3). Firstly, the average Si foliar stock (Fig. 4b) reached ~1000 mg m$^{-2}$ at the Minimal area,

whereas it decreased to ~635 mg m$^{-2}$ at the Moderate area and ~740 mg m$^{-2}$ at the Extensive area. The species contribution to foliar Si stocks varied widely across the thermokarst gradient, according to the species biomass composition and their specific



Si foliar concentrations. Mosses and sedges having high Si (foliar) concentration, their respective contribution to Si foliar stocks was mostly biomass-driven. At Minimal and Extensive areas, mosses showed the highest contribution to foliar Si stocks (~50-60%) because of their high biomass, whereas sedges showed a lower contribution to Si foliar stocks (~25%). Conversely,

Moderate area displayed particularly high foliar biomass of *E. vaginatum* (as part of sedges) and a lower moss biomass, resulting in a higher contribution of sedges than mosses to the Si foliar stock (64% and 20%, respectively). Overall, shrubs and forbs contributed to 5-10% of the Si foliar stock. Secondly, Ca and Mn foliar stocks showed a common trend in foliar stock fluctuation between the thermokarst areas, with Minimal and Extensive areas showing similar Ca and Mn foliar stocks, and Moderate area showing slightly lower Ca and Mn foliar stocks (Fig. 4d-f). Despite the non-significant ($p > 0.05$)

differences between the total foliar stocks of Ca and Mn across the thaw gradient areas, the high Ca and Mn concentrations in shrub and forb foliage (Fig. 2c) resulted in much larger contributions of shrubs and forbs to Ca and Mn foliar stocks (~40% at Minimal, ~50% at Moderate, and ~60% at Extensive area), compared to Si foliar stocks. Consequently, species biomass distribution and foliar elemental composition conjointly influenced the site-specific total mineral element stocks, and the relative contributions of vegetation species to these foliar stocks.

Across Gradient site, vegetation biomass distribution can be explained by the thermokarst development and the induced changes in soil conditions (Schuur et al., 2009). Drier soil conditions upon thermokarst development generated a shift in vegetation towards shrub expansion (Jasinski et al., 2018), where the shrub and forb aboveground biomass accounted for ~40% at the Minimal area and up to ~60% at the Extensive area. Consequently, mineral elements highly concentrated in shrub and forb leaves, such as Ca and Mn (Fig. 2c), have seen their respective shrub and forb foliar stocks increase upon the early

shrubification occurring at Gradient site. Upon more advanced shrubification, we can expect a significant increase in the Ca and Mn total foliar stocks, led by the shrub foliar contribution. The large standard deviations for the mineral element foliar stocks at Gradient site arose from the large heterogeneity in species foliar biomass, with relative standard deviation between 2- and 10-times higher for foliar biomasses (Table S.3) than for foliar concentrations (Mauclet et al., 2021b).

### 3.3.2 Influence of permafrost degradation on potential mineral element foliar fluxes

Shifts in mineral element foliar stocks upon shrubification may directly affect annual foliar fluxes from plant leaves to soil. To assess the influence of drier soil conditions (upon thermokarst process) on the mineral element cycling, we focussed on the five vascular species subjected to seasonal senescence and litterfall (*E. vaginatum*, *C. bigelowii*, *R. chamaemorus*, *V. uliginosum*, and *B. nana*). We did not include evergreen species, nor mosses and lichens.

At the least degraded area (Minimal) of Gradient site, mineral element annual foliar fluxes ranged from 670 mg m$^{-2}$ a$^{-1}$ for K

to 9-6 mg m$^{-2}$ a$^{-1}$ for Fe and Zn (Fig. 5b). Depending on the mineral element, the species contribution to annual foliar fluxes varied greatly, but sedge contribution remained important to every annual foliar flux (between 40% for Ca and 90% for Si foliar flux). Considering the thermokarst gradient (i.e., Minimal, Moderate, and Extensive areas), overall differences in mineral element foliar fluxes across gradient sites were biomass-driven. Firstly, total annual Si foliar fluxes were significantly different ($p < 0.05$), and varied from 390 mg m$^{-2}$ a$^{-1}$ at Moderate area to 220 mg m$^{-2}$ a$^{-1}$ at Extensive (Fig. 6b). Moderate area showed





the highest total annual Si foliar flux due to high biomass and productivity of Si-rich plant species (such as sedges), whereas the Extensive area showed the lowest total annual Si foliar flux due to high biomass and productivity of Si-poor plant species (such as shrubs). Similar to Si, other mineral elements such as K, Fe, and Zn also showed a significant decrease ($p < 0.05$) in their annual foliar fluxes between Moderate and Extensive areas (30 to 40% decrease), i.e., upon thermokarst development and early shrubification (Fig. S.5). Secondly, in contrast to the net decrease in annual Si, K, Fe, and Zn foliar fluxes, Ca and

Mn showed similar annual foliar fluxes across the thermokarst gradient sites, with an increasing contribution of shrubs and forbs to these foliar fluxes: from ~45% at Moderate to ~75% at Extensive (Fig. 6d-f). For Ca and Mn, the annual foliar fluxes were positively affected by the increase in shrub biomass and productivity observed at the Extensive area. More specifically, deciduous shrubs and forbs showed increasing annual Ca and Mn foliar fluxes upon permafrost degradation and the induced shrubification (Mauclet et al., 2021b), therefore offsetting the simultaneous decrease in sedge contribution. It should be noticed

that elemental foliar stocks and fluxes at Gradient have been estimated with the foliar concentrations measured on leaf sampled in August 2019, and the available foliar biomass and productivity data from July 2017. Although this may add a cofounding factor, we have seen that foliar elemental composition was specific to vegetation functional types (section 3.1), well in line with other studies (Aerts et al., 1999; Hobbie and Gough, 2002; Shaver and Chapin, 1991; Urbina et al., 2017).

At Gradient site, where permafrost has been thawing over the past decades and thermokarst has developed (Schuur et al.,

2008), differences in vegetation community biomasses suggest that shrubification occurs upon permafrost degradation (Jasinski et al., 2018). Foliar productivity followed the same pattern as foliar biomass, with sedges dominating plant productivity at the Minimal and Moderate areas, and shrub dominating plant productivity at the Extensive area. We emphasize that shifts in vegetation biomass distribution and productivity upon shrubification led to an increase in foliar fluxes of mineral elements relatively more concentrated in shrub than sedge foliar tissues (i.e., Ca and Mn), and to a decrease in mineral elements

relatively less concentrated in shrub than sedge foliar tissues (i.e., Si, K, Fe, and Zn; Fig. 2c).





**Figure 3: (a) Cumulative mineral element stock (mg m⁻²) into foliar tissues of five vascular plant species, at the Control site of set up for the Carbon in Permafrost Experimental Heating Research (CiPEHR) in 2009 and 2017. Elemental content of mosses and lichens was not measured at the CiPEHR site. Letters correspond to a mixed model analysis and compare the total foliar elemental stocks between 2009 and 2017. (b) Cumulative mineral element stock (mg m⁻²) into aboveground tissues of vascular and non-vascular species of the least degraded area (Minimal) of the thermokarst gradient (Gradient site) in 2017. Vegetation species are sorted by plant functional types: deciduous shrubs and forbs (green), evergreen shrubs (blue), sedges (yellow), lichens and mosses (red). Error bars represent standard deviations.**






**Figure 4: (a-c-e)** Cumulative mineral element stock (mg m$^{-2}$) into foliar tissues of five vascular plant species at the Carbon in Permafrost Experimental Heating Research (CiPEHR) in 2009 and 2017. Elemental content of mosses and lichens was not measured at the CiPEHR site. The four treatments (Control, Summer warming, Winter warming, and Annual warming) respectively refer to no artificial treatment, air warming, soil warming and both (air and soil) warming. Letters correspond to a mixed model analysis and compare the total foliar elemental stocks between warming treatments and years. **(b-d-f)** Cumulative mineral element stock (mg m$^{-2}$) into aboveground tissues of vascular and non-vascular at the thermokarst gradient (Gradient site) in 2017. Stages of permafrost degradation are classified as Minimal, Moderate, and Extensive, based on permafrost thaw and subsidence rate. Letters correspond to a one-way ANOVA test and compare the total foliar elemental stocks between the three stages of permafrost degradation. Vegetation species are sorted by plant functional types: deciduous shrubs and forbs (green), evergreen shrubs (blue), sedges (yellow), lichens and mosses (red). Error bars represent standard deviations.







**Figure 5: (a) Cumulative annual foliar flux (mg m⁻² a⁻¹) of mineral elements for the considered species at the Control site of set up for the Carbon in Permafrost Experimental Heating Research (CiPEHR site) in 2009 and 2017. Letters correspond to a mixed model analysis and compare the total foliar elemental stocks between 2009 and 2017. (b) Cumulative annual foliar flux (mg m⁻² a⁻¹) of mineral elements for the considered species at the least degraded area (Minimal) of the thermokarst gradient (Gradient site) in 2017. Vegetation species are sorted by plant functional types; deciduous shrubs and forbs (green), and sedges (yellow). Error bars represent standard deviations.**



**Figure 6: Cumulative annual foliar flux (mg m⁻² a⁻¹) of mineral elements for the considered species at (a-c-e) the Carbon in Permafrost Experimental Heating Research (CiPEHR) in 2009 and 2017, and (b-d-f) the thermokarst gradient (Gradient site) in 2017. At CiPEHR, the four treatments (Control, Summer warming, Winter warming, and Annual warming) respectively refer to no artificial treatment, air warming, soil warming and both (air and soil) warming. Letters correspond to a mixed model analysis and compare total foliar elemental fluxes between warming treatments and years. At Gradient site, stages of permafrost degradation are classified as Minimal, Moderate, and Extensive, based on permafrost thaw and subsidence rate. Letters correspond to a one-way ANOVA test and compare total foliar elemental fluxes between the three stages of permafrost degradation. Vegetation species are sorted by plant functional types: deciduous shrubs and forbs (green), and sedges (yellow). Error bars represent standard deviations.**





### 3.4 Implications for vegetation shifts in the Arctic and sub-Arctic

Vegetation communities across the Arctic and sub-Arctic are sensitive to a warming environment, as supported by many field studies (Chapin and Shaver, 1985; Shaver, 1996; Viers et al., 2013; Wookey et al., 2009) and satellite observations (Bhatt et

al., 2017; Myneni et al., 1997; Pouliot et al., 2009; Xu et al., 2013). Upon changing environmental conditions (i.e., air temperature, growing season length, snow cover, and soil moisture regime), shifts in plant functional group dominance may spatially differ (i.e., shrub or sedge dominance) and have contrasting implications on the surrounding environment. For instance, local differences in vegetation cover may have effects on heat transfer, shading, snow-trapping, and solar radiation reflectivity (e.g., Chapin et al., 2005; Sturm et al., 2001a). In this section, we discuss the influence of vegetation community

structure and composition on mineral element cycling at a larger scale of the Arctic upon: (i) shrubification, (ii) graminoid expansion, and (iii) changes in non-vascular species composition.

A major observation across the Arctic tundra is the expansion of woody shrubs, in terms of biomass, cover, and height (Forbes et al., 2010; Hobbie and Chapin, 1998; Mod and Luoto, 2016; Myers-Smith and Hik, 2018; Stow et al., 2004; Sturm et al., 2005, 2001; Tape et al., 2006; Walker et al., 2006). While shrubification seems to have started at the close of the Little Ice

Age (ca. mid-19th century) and have continued with accelerating warming in the latter half of the 20th century (Tape et al., 2006), it is predicted to increase by as much as 52% by 2050 (Pearson et al., 2013). Shrubification has multiple implications on surrounding ecosystems and may have positive feedback effects on warming and shrub promotion (Mod and Luoto, 2016; Weintraub and Schimel, 2005), e.g., by modifying the surface energy budget through change in albedo and soil surface roughness (Chapin et al., 2000; Sturm et al., 2001b). Furthermore, Arctic shrubification may affect biogeochemical cycles, by

potentially increasing total annual foliar fluxes of mineral elements highly concentrated in shrubs (such as Ca and Mn; Fig. 2c), or decreasing total annual foliar fluxes of mineral elements poorly concentrated in shrubs (such as Si, K, Fe, and Zn; Fig. 2c). While Ca is a non-limiting macronutrient for plants, Mn is a micronutrient sensitive to soil redox conditions that can influence the organic matter decomposition (Jones et al., 2020; Keiluweit et al., 2015). Soil saturation regime (Herndon et al., 2020) and ability of plants and microorganisms to supply, accumulate, and regenerate short-lived $Mn^{3+}$ species in the litter

(Keiluweit et al., 2015) govern the forms of Mn present in the soil, which in turn affect litter decomposition efficiency. Upon Arctic shrubification, the potential increase in foliar Mn fluxes from plant to soil surface could therefore promote litter decomposition rate, a key factor in nutrient cycling and availability, plant growth, and terrestrial C balance, thereby contributing to amplify the permafrost C feedback (Schuur et al., 2015).

A parallel change in tundra vegetation species abundance is the expansion of sedges (as part of the graminoids) at localized

sites across the Arctic and sub-Arctic regions, including CiPEHR at EML. At numerous warming sites, an increase in sedges (as *E. vaginatum*) occurred at the expense of shrubs during abrupt permafrost thaw, with pond development and wetter soil conditions (van der Kolk et al., 2016). While shrub growth can be limited by very wet soil conditions and low nutrient supply, sedges can develop in a wide range of soil moisture conditions thanks to their deeper rooting systems, and are able to reach underlying soil layers and access deeper nutrient pools. At CiPEHR, the evolution of mineral element foliar stocks (Fig. 4a-c-



e) and potential fluxes (Fig. 6a-c-e) with time demonstrated that nutrient uptake promoted by sedges (such as Si, P, and Fe) may positively influence the global mineral element storage in tundra plant tissues, and thereby the transfer of mineral elements from plant leaves to soil litter. For example, we observed large increase in Si, P, and Fe foliar stocks (Fig. 4a and Fig. S.2) and potential annual foliar fluxes (Fig. 6a and Fig. S.4), upon sedge expansion. While Si and Fe are non-essential plant nutrients, P is a limiting macronutrient for the vegetation (DalCorso et al., 2014; Marschner, 2012; Schachtman et al., 1998). In moist

tundra ecosystems, P input into organic surface soils is small, and P recycling from organic soils supplies most of the P taken up by plants (Shaver et al., 1991). Therefore, a potential increase of P input in the upper soil layers through higher annual P foliar fluxes may positively influence the global vegetation productivity. However, P mineralisation and availability may not simultaneously increase upon warmer soil conditions. Competition between vegetation and microbial demand for mineral nutrients can severely limit plant P availability in Arctic soils (Jonasson et al., 1993; Nadelhoffer et al., 1991). According to

the litter decomposition rate upon stimulated microbial activity, P bioavailability may evolve and directly influence vegetation growth and development. As vegetation is a key element of the total C budget, an increase in the cycling rates of limiting nutrients such as P may indirectly influence the role of vegetation in C sequestration through higher plant development and photosynthesis.

Finally, non-vascular species, such as mosses and lichens, dominate the groundcover biomass and productivity of many

northern tundra ecosystems (Beringer et al., 2001; Longton, 1988; Oechel and Van Cleve, 1986; Viereck et al., 1986). Mosses display physiological and ecological traits that greatly influence soil thermal (Luthin and Guymon, 1974) and hydrological regimes (Zimov et al., 1995), but also plant nutrient availability (Cornelissen et al., 2007; Turetsky, 2003; Turetsky et al., 2010). Our low-scale study at Gradient site highlighted the important moss contribution to mineral element foliar stocks (Fig. 3b and Fig. 4b-d-f). Besides the important variability in moss biomass composition at Gradient site (with coefficients of

variation ranging from 80% at Extensive area, up to more than 100% at Minimal area, data from Jasinski et al., 2018), mosses showed particularly high concentration in mineral elements such as Si, Al and Fe (Fig. 2c, with coefficients of variation ranging around 24%). Therefore, besides affecting local to regional environmental soil conditions (temperature and moisture), changes in moss biomass composition may also affect nutrient cycling and availability, and therefore future patterns of vegetation shift (Beringer et al., 2001).

## 4 Conclusion

In this study, we tested the influence of permafrost degradation on the mineral element foliar stocks and potential litterfall fluxes, upon two different shifts in sub-Arctic tundra vegetation. First, we measured the foliar elemental composition of five vascular plant species (part of graminoids, deciduous and evergreen shrubs, and forbs) at an experimental permafrost warming site (CiPEHR) undergoing a sedge (as part of graminoid) expansion upon wetter soil conditions. We also measured the foliar

elemental composition of seven vascular plant species (part of graminoids, deciduous and evergreen shrubs, and forbs) and two groups of non-vascular species (mosses and lichens) at a natural thermokarst gradient site (Gradient) facing shrubification



upon permafrost degradation and drier soil conditions. Our results highlighted the specificity of vegetation functional types in their foliar elemental composition, according to their physiological, phenological, and ecological properties. Sedges (i.e., *E. vaginatum* and *C. bigelowii*) showed relatively high foliar Si, P, and Fe concentration, whereas shrubs (*V. uliginosum*, *B. nana*, *V. vitis-idaea*, and *R. tomentosum*) showed relatively high foliar Ca and Mn concentration. The main conclusions are:

(i) In cases of wetter soil conditions and sedge expansion (or more largely graminoid expansion), mineral elements that were highly concentrated in sedge foliar tissues (such as Si, P, and Fe) showed large increase in their foliar stocks. This results mainly from the increase in sedge biomass across the site (CiPEHR). In parallel, the higher sedge foliar productivity generated higher annual maximum potential foliar transfers from plant to soil litter for those mineral elements (considering that fluxes of elements such as P are overestimates).

(ii) In cases of drier soil conditions and shrubification, mineral elements highly concentrated in shrub foliar tissues (such as Ca and Mn) would increase in their total foliar stocks. This results mainly from the increase in the shrub contribution to their respective foliar stocks, upon the early shrubification occurring across the thermokarst gradient site (Gradient). We expect a larger increase in the total foliar Ca and Mn stocks, upon a wider shrubification. Similarly, the increasing shrub foliar productivity upon permafrost degradation promoted annual fluxes of Ca and Mn from plant to soil litter.

(iii) The spatial heterogeneity in permafrost soil degradation and soil moisture conditions is leading to contrasting shifts in vegetation (graminoid and shrub expansion) across the Arctic. We observed that mineral element cycling directly depends on the vegetation biomass and diversity. We therefore suggest that the different plant functional groups should be considered in future studies, in order to allow more accurate predictions on the influence of vegetation shifts on biogeochemical cycling of the elements in Arctic ecosystems.

Here, we provided first estimations of potential mineral element fluxes from plant to soil, via annual leaf fall, and we highlighted that shifts in vegetation (i.e., sedge or shrub expansion) influence the mineral element composition of the litter. In the future, there is a need to assess to what extent the shift in vegetation and thereby in litter elemental composition will accelerate or slow down the litter decomposition, and how fast these mineral elements may be available again for vegetation uptake. This would require integrating rates of litter decomposition and mineralisation through microbial activity with local shifts in vegetation biomass and productivity.

**Data availability**

Elemental foliar stocks and potential litterfall fluxes presented in this paper are available in the Bonanza Creek LTER (CiPEHR dataset: https://doi.org/10.6073/pasta/597c40c5d699eec918da3e9c2eaa7bea, Mauclet et al. 2021a; and Gradient dataset: https://doi.org/10.6073/pasta/7fad9398ec3a596b8efc092fc8fbf55d, Mauclet et al. 2021b).





**Supplement**

*See Supplement*

**Team list**

Elisabeth Mauclet[1*], Yannick Agnan[1], Catherine Hirst[1], Arthur Monhonval[1], Benoît Pereira[1], Aubry Vandeuren[1], Maëlle Villani[1], Justin Ledman[2], Meghan Taylor[2], Briana L. Jasinski[2], Edward A. G. Schuur[2], Sophie Opfergelt[1]

[1]Earth and Life Institute, Université catholique de Louvain, Louvain-la-Neuve, Belgium
[2]Center for Ecosystem Science and Society, Northern Arizona University, Flagstaff, AZ, USA

**Author contribution**

EM, CH, AM and SO planned the campaign. SO obtained the grant that funded this research. JL, MT, BJ, and EAGS provided great help for the field work, supplied foliar samples, and shared massive datasets. EM, CH, AM, and MV performed the measurements. EM, YA, and SO analyzed the data. BN and AV helped with statistical analyses. EM wrote the manuscript. CH, SO, YA, MT, and EAGS reviewed and edited the manuscript. All authors gave final approval for publication.

**Competing interests**

The authors declare that they have no conflict of interest.

**Acknowledgements**

We warmly acknowledge the master students who participated to the data collection: Loïc Debry, Laurentine Debruxelles and Simon Malvaux. Thanks to Anne Iserentant, Hélène Dailly, and Elodie Devos from the MOCA analytical platform at
UCLouvain for mineral elemental analyses, and to the SMCS platform from the UCLouvain for the statistical support. We also greatly thank the Schuur Lab (Northern Arizona University) for their scientific support and the sample collection.

**Funding**

This work was supported by the European Union's Horizon 2020 research and innovation program (grant agreement No. 714617, 2017-2022) and by the Fund for Scientific Research FNRS in Belgium to SO (FC69480).





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
