# Peer review of "Changing sub-Arctic tundra vegetation upon permafrost degradation: impact on foliar mineral element cycling"

_Biogeosciences, 2021_

## Author Comment (AC1)

**Responses to the comments of Jonathan von Oppen on the manuscript 'Changing sub-Arctic tundra vegetation upon permafrost degradation: impact on foliar mineral element cycling' (Biogeosciences discussion BG-2021-263)**

Mauclet et al. present leaf mineral micronutrient analyses from two arctic tundra sites, in the light of potential vegetation changes with ongoing permafrost degradation. The sites represent one experimental and one observational system, with contrasting vegetation dynamics upon soil permafrost thawing. The authors convert measured leaf nutrient contents into foliar stocks and maximum potential foliar fluxes to assess the effect of potential vegetation shifts on future nutrient cycling.

The study is timely and thoroughly conducted. The authors well explain the general study background and give a detailed description of the methods and results. The methodology appears overall sound and well justified, with some minor inaccuracies as outlined below, and the authors discuss some important implications of their findings.

We thank the reviewer Jonathan von Oppen for his careful reading and the constructive comments that will improve the quality and the readability of the manuscript. We are happy to apply revisions to improve our manuscript as formulated in the answers to the referee comments.

However, from my perspective, the study does not fulfil its potential, as the extensive amount of detail given as well as a confusing structure make it difficult for the reader to extract the most important messages. This includes, but is not limited to (i) repetition of methodological details that could be avoided if it had been explained more clearly in the first place; (ii) inconsistencies of figure contents vs. order of paragraphs in the manuscript, resulting in thematic jumps; (iii) the lack of a clear line of argument within most paragraphs and sections; (iv) complex sentence structure; and (v) language errors and unfitting use of expressions (while I acknowledge that English might not be the lead author's first language).

Response: We have addressed this comment by revising the structure of the manuscript. In the new version, (i) we have better constrained the methodological details in the Section Material and Method and cleaned the excessive methodological information in the Section Results and discussion, (ii) we have split the figures to ensure consistency between paragraphs and figure contents and avoid thematic jumps, (iii, iv) we have rephrased complex sentences with shorter and clearer line of arguments, and (v) we have carefully checked language errors and expressions.

These drawbacks also impede a proper evaluation of technical and argumentative details, particularly in the very long results/discussions section (and especially 3.1 – 3.3).

Response: We agree with the reviewer. We carefully revised the text to remove excessive information and re-arranged the long paragraphs to be more concise and clearer in the argumentation.

I would therefore suggest thorough revision of the manuscript with the aim of a more concise language, including considerations of which details could be left out, and the establishment of a clear red line of argument. One potential way to achieve this could be a more hypothesis-based structure. Clearly reasoning and stating the expected findings at the outset would help the reader to follow the line of argument, and could provide the authors with a consistent structure to follow for the rest of the manuscript. As much of the confusion arises from parallel evaluation of the two different study sites, the authors might also consider focusing on either the experimental or the natural gradient site.

Response: We thank the reviewer for this comment. As stated above, we carefully revised the text to remove excessive information and re-arranged the long paragraphs to be more concise and clearer in the argumentation. We understand that the structure of the manuscript was a source of confusion to grasp the main messages from the parallel evaluation of the experimental and the natural gradient site. Thanks to the comment of the reviewer, we have modified the way to introduce the experimental and the natural gradient site and their specific importance for the study. We have clarified that we rely on CiPEHR as a study case of graminoid expansion in wetter soil conditions, and Gradient as a study case for shrubification in drier soil conditions. Focusing on either the experimental or the natural gradient site would be a loss given that these two approaches bring complementary information to convey the main message of the manuscript. We agree that a new structure was necessary in order to keep the two study sites.

In summary, I do think this study has the potential to make a valuable contribution for Arctic research eventually. However, I encourage the authors to extensively and carefully revise the text to help guiding the reader and support the transfer of their findings and implications.

Below I am listing some detailed comments that I hope will also be relevant and helpful for a revised version:

*Introduction*:

The introduction is generally well structured, but contains much detail that is perhaps not necessary. I recommend starting each paragraph with a clear statement of what information it contains, and wrapping each paragraph up with a summarising statement which supports transition into the next point.

As mentioned before, I suggest ending the introduction with a clear hypotheses statement as a guideline for the reader and to help structure the text later on.

Response: We addressed this comment by removing the excessive information and re-organizing the paragraph structure of the Introduction. We also clarified the objectives with a hypothesis statement at the end of the introductive section, L 80-86.

*Material and Methods*:

The methods section gets very complex by the methodological differences between the gradient and the experimental sites. This applies especially to the "Data treatment" section. As I suggested above, focusing on one of the two sites could help to reduce complexity.

Response: We addressed this comment by reformulating and simplifying the explanations in the "Data treatment" section. This section is essential to explain the calculations. As explained here above as a response to comment 3, both sites are necessary, and we have improved to structure for clarity.

The methods contain a detailed description of the CiPEHR treatments, yet I don't see the effects of these being evaluated in detail in the results and discussion. I suggest to either discuss the effects of the different warming treatments, or to remove unnecessary detail from the methods description.

Response: We have simplified the details presented for the CiPEHR treatments and removed the unnecessary details from the Method section. We discuss the effects of the warming treatment at CiPEHR (L259-265; L323-328). We explain that these treatments do not have a significant effect on the elemental foliar stocks and fluxes, which are mainly controlled by the sedge expansion over time. Therefore, the discussion about the effects of these warming treatments is now brief.

L259-265: "The effect of the four artificial warming treatments on foliar stocks was evaluated in 2009 and in 2017 for Si, Ca, and Mn (Fig. 4), and for K, Zn, S, P, Fe, and Al (Fig. S.2). In 2009 as in 2017, mineral element foliar stocks showed no significant difference ($p > 0.05$) between warming treatments and control sites. Despite the increasing degree of permafrost degradation between the four warming treatments in terms of subsidence and rising water table (control < summer warming < winter warming < annual warming; Rodenhizer et al., 2020), only time had a positive effect on mineral element foliar stocks with total foliar stocks being 1.5- to 5-times higher in 2017 than in 2009 ($p < 0.05$, for almost every mineral element, except for Ca and Mn)."

L323-328: "The effect of the four artificial warming treatments on the potential annual foliar fluxes was evaluated in 2009 and in 2017 for Si, Ca, and Mn (Fig. 6), and for K, Zn, S, P, Fe, and Al (Fig. S.4). In 2009 as in 2017, some mineral elements (Ca and Mn: Fig. 6b-c; K, Fe, S, and Zn: Fig. S.4) showed differences in their potential annual foliar fluxes with warming treatments. However, this variability in the potential foliar fluxes between the four treatments was much less pronounced than the large increase in the potential elemental foliar fluxes that occurred overtime, with total foliar fluxes 1.5- to 10-times higher in 2017 than in 2009 ($p < 0.05$, Table S.5)."

If keeping the details on treatments, presenting data on their effectiveness (e.g., soil temperatures across warmed seasons) in the supplementary material could help the reader assess treatment effects on the vegetation. The same goes for vegetation biomass data: even if that has been published previously, it is very central to the argumentation and interpretation of the results of this study.

Response: As explained in the response to the previous comment, these treatments do not have a significant effect on the elemental foliar stocks and fluxes, which are mainly controlled by the sedge expansion over time. Therefore, presenting additional details about those warming treatments would be confusing. Following the suggestion of the reviewer, foliar biomasses used for the stock and flux estimations are included in the Supplementary material (Table S. 3).

I could not find any reasoning why the number of species sampled differed between the gradient and the experimental site.

Response: On one hand, vegetation samples from CiPEHR were collected and provided by researchers from the NAU (Northern Arizona University, USA). We benefited from this collaboration to evaluate the influence of 8-years of warming on vegetation elemental composition. The plant species provided did not include *Rhododendron subarcticum*, *Carex bigelowii*, and the non-vascular species (mosses and lichens) were not collected during this experiment. On the other hand, vegetation samples from Gradient were collected more recently as part of the research project WeThaw at UCLouvain (Belgium), and we had the opportunity to collect more plant species. This is the reason why more plant species are available in Gradient. We have modified the sentence in the method section to clarify that "At CiPEHR site, we collected five of the most abundant vascular species" (Line 139), and that "At Gradient, we were able to collect seven of the most abundant vascular species" (Line 144).

Likewise, I suggest to assist the reader with understanding of where the number of leaf samples arises from (L166). I cannot see a clear relationship with the number of sites or species.

Response:  A full description of each number of leaf samples by species, site, area/treatment, and year (for CiPEHR) is presented in the Suppl. Mat. (Table S.1). For clarity, we have included the numbers of samples in the "Sampling method" section (L141 and L146).

I am missing an evaluation of the model assumptions for applying a parametric test. Homogeneity of variances is briefly mentioned in the supplementary material, but does not receive consideration in the main text.

Response: This has been clarified in the revised manuscript (L205-215).

L229: how could an interaction between treatments be included in the model if "treatment" represents a single factor?

Response: There was a wrongly stated explanation at L219 that has been corrected in the revised version of the manuscript (L.205-215). In the statistical analysis, we used mixed-effects models to investigate separately the effects of experimental warming overtime on the mineral element foliar stocks and fluxes (Table S.5). The mixed-effects models included a random effect for repeated measurements on individual plots and the plot-level foliar stocks and fluxes were used as dependent variable. The model included the treatments (summer, winter, and annual warming) and the time as covariates, and the interaction between treatments and time.

Numbering of the Methods subsections should be 2.x instead of 1.x

Response: This has been corrected and adapted across the manuscript.

Figure 1 is currently not referenced in the text.

Response: This has been corrected (L.106)

*Results and Discussion*:

Though I like the use of numbers in the text, I would recommend to not present every detailed result for every species and response. This makes the reader tired and distracts from the main messages. Instead, I would recommend to focus on the most important findings and the overall picture for every analysis.

Response: We thank the reviewer for the comment. Excessive details have been removed from the main text of the discussion.

I would suggest to follow the structure One figure, one paragraph/section – don't come back to different panels later on (e.g. Fig. 3 and sections 3.2.1 and 3.3.1). Instead, group both paragraphs and figures thematically and consistently.

Response: We changed the arrangement of the figures, in order to better correspond to the paragraph description.

Alternatively, as mentioned above, focusing on either the gradient or the experimental site might help with streamlining the text.

The intermingled structure of results and discussion in sections 3.1 to 3.3 makes it hard to distinguish descriptions of results from discussing statements, especially as the authors frequently refer to detailed results from other publications. In such cases, I suggest separating results and discussion into consecutive paragraphs.

Response: This has been addressed along the manuscript; we have separated results and discussion in different paragraphs.

I suggest to carefully consider the importance of individual findings. Presenting the most important results first will help the reader to extract information.

I had the impression that the depth of discussion was not well balanced across all findings. Some results were discussed in much detail and with a lot of references (e.g. section 3.1), while others were described in great detail, but only briefly evaluated (e.g. section 3.2.2). In both cases, focusing more on the most important aspects for the general picture might help in creating that balance.

Response: According to the earlier comments of the reviewer, the general structure has been improved for clarity and the balance between results and discussion has been carefully checked.

I would also recommend not to bring up methodological details in the discussion again – for instance L270f, "In 2009 (before any experimental warming), as in 2017 (after eight years of experimental warming)". If clearly described in the Methods section, this will not be necessary.

Response: This has been addressed along the Results and discussion section.

The last, synthesising part of the discussion reads well! I think it would benefit from making it more concise and to the point as well, but it sums up the study implications nicely.

Response: We thank the reviewer, we have made it more concise as suggested.

The same goes for the conclusion, which should also be shortened considerably to emphasise the most important points. There's no need to recap the methodology in such a detailed way here, that only takes the focus of the reader away from the important outcomes.

Response: We thank the reviewer, the conclusion was revised and shortened.

*Language remarks*:

I suggest to carefully revisit the use of "the" throughout the manuscript. For instance, it should be used when referring to specific subsites ("at the Gradient site").

Response: This has been corrected throughout the manuscript.

Similarly, I would advise not to use commas excessively as they tend to break the reading flow. (There are some good overviews of comma rules available online, such as https://owl.purdue.edu/owl/general_writing/punctuation/commas/extended_rules_for_commas.html)

Response: We thank the reviewer for the advices, the use of comas has been reduced through the manuscript.

I also recommend to refer to the degradation stages of the gradient site as e.g. "the Moderate thaw area" instead of just "Moderate area" throughout the manuscript to enhance clarity.

Response: This has been corrected across the manuscript.

To my knowledge, the expression "vegetation species" does not exist in English, and it might even create confusion in whether the authors are referring to the vegetation as the entity of species or communities across an area, or to separate species as such. Assuming the latter, I think that using "plant species" or simply "species" should do in most instances.

Response: We had used the expression "vegetation species" because it includes the moss and lichen species, unlike "plant species" that is more restrictive and might lead to confusion. To address this

comment, we have modified the expression "vegetation species" with an adapted terminology ("species" or "plant species" when appropriate).

If using an expression like "A was explained by B", I would expect that this relationship was by some means statistically tested. If, like for instance in L395, referring to a matching pattern or logical reason for a finding, I would rather phrase this like "A mirrored B" or "A followed B" to avoid confusion. But that might be a matter of taste :)

Response: We thank the reviewer for these suggestions which have been included throughout the manuscript.

Other minor points:

The authors might want to check consistency of reference formatting. E.g. L479 "Chapin et al. 2005" vs. L373 "Chapin III et al. 1980".

Response: We thank the reviewer for this remark. This has been corrected throughout the manuscript and in the references.

As the text will need thorough revision throughout, I hope the authors and editor will understand that I don't give more detailed comments on phrasing of specific passages at this stage. I will be happy to provide these once the authors have revised the overall structure of the manuscript.

Response: We thanks again Jonathan von Oppen for his time and valuable comments on our work.

---

## Referee Report (RR1)

I would like to congratulate the authors on the improved revised manuscript. Readability has benefitted tremendously from restructuring, especially in Results and Discussions section. Well done! Conciseness also more concise now, though I see some potential to condense even more (e.g., in the very extensive study site description, or the summary ("Overall, …") after the summary ("In summary, …") in the two lower paragraphs on p. 13). I have included some additional suggestions for changes below, but would be happy to see the manuscript published after consideration of these points.

Materials and Methods

L141/146: is the number of samples equal to the number of sampled leaves *per species*? I suggest to explain what exactly was sampled (as in L142f). For example, if $n = 69$ is the total number of samples at the Gradient, Fig. 2c wouldn't make sense as it has more levels (81).

L176ff: the $\Delta$ symbol is normally used for differences, not standard deviations, so I find its use here a bit confusing-.

L205: which variables were log-transformed?

L207ff: including model formulas could help with clarity.

Results/Discussion

L225f: the statement "potentially induced by…" is repeated in the following paragraph. It seems more meaningful there, so I suggest to remove it here.

L248f: I don't think it's necessary to cite the same reference twice within this sentence and suggest to remove the first instance.

Fig. 5/6: I think the captions of the two figures are interchanged. Further, both would benefit from a more detailed description of what they are displaying in the first sentence, e.g. "foliar fluxes across treatments" (Fig. 5) vs. "between treatments" (Fig. 6).

L. 481: "The wide shrubification across the Arctic is expected to increase by as much as 52% by 2050 (Pearson et al., 2013)." This is a very vague statement. For instance, what baseline is that increase compared to? Are you referring to increases in shrub cover and/or biomass? I would advise to carefully formulate such general statements to still keep them precise (and thus impactful).

L. 483: to my understanding, Mod and Luoto (2016) neither discuss albedo nor soil surface effects of shrubification. Consider removing this reference.

Conclusion

L528ff: I suggest to make it clear that conclusions (i) and two (ii) refer to the community level, in contrast to the initial concluding statement at species level.

L548ff: I find the last two sentences a bit confusing. After only discussing micronutrients throughout the discussion, this reference to carbon cycling distracts from the main take-home message of the paper. If you want to include this point, I suggest to move it to the previous section (Implications for vegetation shifts).

Additional remark:

I would like to reiterate my suggestion of phrasing references to degradation stages at the Gradient site as "Minimal *thaw* area" instead of just the "Minimal area" (e.g. L519), and likewise for the other two degradation levels, to improve clarity for the readers.

---

## Author Response (AR2)

*Responses to reviewer*

**Changing sub-Arctic tundra vegetation upon permafrost degradation: impact on foliar mineral element cycling**

I would like to congratulate the authors on the improved revised manuscript. Readability has benefitted tremendously from restructuring, especially in Results and Discussions section. Well done!

We thank the reviewer Jonathan von Oppen for his second careful reading of the document and helpful comments.

Conciseness also more concise now, though I see some potential to condense even more (e.g., in the very extensive study site description, or the summary ("Overall, …") after the summary ("In summary, …") in the two lower paragraphs on p. 13). I have included some additional suggestions for changes below, but would be happy to see the manuscript published after consideration of these points.

We have addressed this comment by revising and simplifying the study site description (section 2.2) and the discussion summary (L290).

Materials and Methods

L141/146: is the number of samples equal to the number of sampled leaves per species? I suggest to explain what exactly was sampled (as in L142f). For example, if n = 69 is the total number of samples at the Gradient, Fig. 2c wouldn't make sense as it has more levels (81).

We have clarified the description of the foliar sampling at the experimental CiPEHR site (L141) and the Gradient site (L150) to specify that "The foliar sampling was performed as one bulk foliar sample per plant species per plot".

Moreover, numbers indicated on the Fig.2a-b-c have been adapted for a better understanding.

L176ff: the ⍰ symbol is normally used for differences, not standard deviations, so I find its use here a bit confusing-.

The symbol Δ has been replaced by the symbol σ according to the comment of the reviewer (L180 and L196).

L205: which variables were log-transformed?

Variables that required log-transformation are indicated in the Suppl. Mat (Table S.5). We have specified this in the text (L206).

L207ff: including model formulas could help with clarity.

This has been included (L217).

Results/Discussion

L225f: the statement "potentially induced by…" is repeated in the following paragraph. It seems more meaningful there, so I suggest to remove it here.

We agree. The first repetition has been removed.

L248f: I don't think it's necessary to cite the same reference twice within this sentence and suggest to remove the first instance.

We agree. The first instance has been removed.

Fig. 5/6: I think the captions of the two figures are interchanged. Further, both would benefit from a more detailed description of what they are displaying in the first sentence, e.g. "foliar fluxes across treatments" (Fig. 5) vs. "between treatments" (Fig. 6).

We agree. This has been corrected now. We also added a more detailed description as suggested (here, and for the similar figures Fig. 3-4-5-6).

L. 481: "The wide shrubification across the Arctic is expected to increase by as much as 52% by 2050 (Pearson et al., 2013)." This is a very vague statement. For instance, what baseline is that increase compared to? Are you referring to increases in shrub cover and/or biomass? I would advise to carefully formulate such general statements to still keep them precise (and thus impactful).

We understand this was not specific enough. We have specified that the increasing shrubification by 2050 was referring to the shrub cover. This has been included L495-496: "The shrub expansion across the Arctic is an important and widely observed response of high latitude ecosystems to rapid climate warming, and the woody shrub cover is projected to increase by as much as 52% by 2050 (Pearson et al., 2013)"

L. 483: to my understanding, Mod and Luoto (2016) neither discuss albedo nor soil surface effects of shrubification. Consider removing this reference.

We agree. References have been adapted. This has been included L.497-498: "In addition to the multiple implications on surrounding ecosystems (e.g., changing the Arctic tundra albedo and soil surface roughness: Chapin et al., 2005; Sturm et al., 2001a; Weintraub and Schimel, 2005)".

Conclusion

L528ff: I suggest to make it clear that conclusions (i) and two (ii) refer to the community level, in contrast to the initial concluding statement at species level.

We thank the reviewer for this great suggestion: we have brought that precision in the conclusions. This has been included L542-545: "Our results at the species level showed that sedges (i.e., *E. vaginatum* and *C. bigelowii*) have relatively high Si, P, and Fe foliar concentrations, whereas shrubs (*V. uliginosum*, *B. nana*,

*V. vitis-idaea*, and *R. tomentosum*) have relatively high Ca and Mn foliar concentrations. As a consequence, the main conclusions for the plant community are: (…)" (L.542-545)

L548ff: I find the last two sentences a bit confusing. After only discussing micronutrients throughout the discussion, this reference to carbon cycling distracts from the main take-home message of the paper. If you want to include this point, I suggest to move it to the previous section (Implications for vegetation shifts).

We agree with this suggestion and the two sentences have been moved earlier in the discussion (section 3.4) (L.488-491).

Additional remark:

I would like to reiterate my suggestion of phrasing references to degradation stages at the Gradient site as "Minimal thaw area" instead of just the "Minimal area" (e.g. L519), and likewise for the other two degradation levels, to improve clarity for the readers.

This has been modified throughout the document.